# Asymmetric interactions between *doublesex* and tissue- and sex-specific target genes mediate sexual dimorphism in beetles

C.C. Ledón-Rettig[1,*], E.E. Zattara[1,*] & A.P. Moczek[1]

Sexual dimorphisms fuel significant intraspecific variation and evolutionary diversification. Yet the developmental-genetic mechanisms underlying sex-specific development remain poorly understood. Here, we focus on the conserved sex-determination gene *doublesex* (*dsx*) and the mechanisms by which it mediates sex-specific development in a horned beetle species by combining systemic *dsx* knockdown, high-throughput sequencing of diverse tissues and a genome-wide analysis of Dsx-binding sites. We find that Dsx regulates sex-biased expression predominantly in males, that Dsx's target repertoires are highly sex- and tissue-specific and that Dsx can exercise its regulatory role via two distinct mechanisms: as a sex-specific modulator by regulating strictly sex-specific targets, or as a switch by regulating the same genes in males and females in opposite directions. More generally, our results suggest Dsx can rapidly acquire new target gene repertoires to accommodate evolutionarily novel traits, evidenced by the large and unique repertoire identified in head horns, a recent morphological innovation.

[1] Department of Biology, Indiana University, 915 E. Third Street, Myers Hall 150, Bloomington, Indiana 47405-7107, USA. * These authors contributed equally to this work. Correspondence and requests for materials should be addressed to C.C.L.-R. (email: crisledo@indiana.edu).

Sexual dimorphisms in morphology, behaviour and physiology are ubiquitous, oftentimes producing phenotypic differences between sexes that far surpass those between species. Further, not all body regions, organs or tissue types within an individual are equally sexually dimorphic. Instead, individuals constitute mosaics of parts that differ in their degree of sexual dimorphism. Differences between sexes, or between tissues within sexes, arise during development, even though genomic content differs little between males and females, or not at all between tissues of the same individual. Thus, differences in gene regulation are generally thought to underlie most sex- and tissue-specific differentiation and much research has therefore sought to characterize regulatory mechanisms that govern sex-specific gene expression. A large number of studies now show that, across metazoans, such expression is frequently directed by *doublesex*/*mab-3*-related genes (*Dmrt* genes)[1], which are characterized by the presence of a highly conserved DNA-binding domain (the DM motif)[2]. One of the best functionally characterized *Dmrt* genes is *doublesex* (*dsx*), which is, uniquely in insects, alternately spliced into male and female isoforms[3]. Although the upstream signalling cascades among insects that connect sex-based chromosomal differences to the differential splicing of *dsx* transcripts are in some cases (but not all[4,5]) well described[6–8], much less is known about the downstream target gene repertoires of Dsx, or how such repertoires might modulate the degree and type of sexually dimorphic development across different body regions or tissue types[9]. In this study, we focused on the interactions between Dsx and its downstream targets in directing sex-specific differentiation across diverse tissues in both male and female beetles.

Primary *dsx* transcripts of all insects studied to date undergo sex-specific splicing to produce isoforms with different oligomerization domains that are therefore capable of interacting with different suites of cofactors[10]. As a result, sex-specific isoforms can modulate the expression of different genes, or the expression of the same genes in different ways, which are then thought to instruct sex-specific development[1]. Evolutionary changes in this target repertoire, and the decoupling of Dsx-mediated gene expression in one tissue type from that of another, then present critical developmental-genetic avenues for the diversification of sexual dimorphisms. However, for the vast majority of taxa and trait types, the size and composition of the target gene repertoire regulated by Dsx is entirely unknown, as is the degree to which target repertoires may be tissue-specific or shared across tissue types. More broadly, exactly how much sex-biased gene expression is actually under the control of Dsx, and whether this applies equally to male and female Dsx isoforms, is largely unknown.

Here we characterize the target repertoire of Dsx across four different tissues of males and females of the bull-headed dung beetle *Onthophagus taurus*, a species whose body regions express various degrees of morphological sexual dimorphism, and which is amenable to both genome-wide transcriptomic profiling and gene function analysis. Specifically, we focused our analysis on four ectodermally derived traits that contribute critically to the development of diverse aspects of sexual dimorphism in this species: (a) brains, which are assumed to have a critical role in enabling the highly divergent behavioural repertoires characteristic of each sex, though little is known about the genetic basis of behaviour in this species; (b) genitalia, which derive from the same larval primordia in each sex yet develop into drastically divergent adult morphologies, as well as diversify rapidly among species[11]; (c) thoracic horns, which are considerably enlarged in males but clearly present in both sexes, and whose function is to aid in the shedding of the larval head capsule during the larval-to-pupal molt[12]; in many species pupal thoracic horns secondarily convert into adult structures used as weapons, though in *O. taurus* both sexes resorb thoracic horns prior to the pupal-to-adult molt[13]; and (d) head horns, which in males manifest in a pair of long curved and exaggerated structures used as weapons in combat with other males, whereas females express a modest ridge in the same region.

Our approach in this study employed the functional knockdown of *O. taurus* male and female *dsx* mRNA isoforms[14] by injection of a double stranded (dsRNA) construct (*OtdsxC*) targeting a region common to all isoforms, followed by genome-wide analyses of gene expression across brains, genitalia, thoracic horns and head horns, and paralleled by an analysis of predicted Dsx-binding sites across the *O. taurus* genome. This approach allowed us to characterize putative Dsx target repertoires in an unbiased fashion, across diverse tissues, and at a genome-wide level. We find that Dsx regulates sex-biased expression predominantly in male tissues, that Dsx's target repertoires are highly sex- and tissue-specific and that Dsx can exercise its regulatory role via two distinct mechanisms: as a sex-specific modulator by regulating strictly sex-specific targets, or as a switch by regulating the same genes in males and females in opposite directions. More broadly, our results suggest Dsx can rapidly acquire new target gene repertoires to accommodate evolutionarily novel traits, evidenced by the large and unique repertoire identified in head horns, a recent morphological innovation. We discuss our results in the light of the different evolutionary histories of male and female Dsx isoforms and the evolutionary lability of sexual dimorphisms across lineages.

## Results

**dsx knockdown phenotypes.** To determine Dsx's role in the post-embryonic development of tissue-specific sexual dimorphism in *O. taurus*, we injected dsRNA derived from a previously developed *OtdsxC* construct[14] into larvae to systemically knock down all known *O. taurus dsx* mRNA isoforms (Fig. 1a), whereas control individuals were injected with non-sense dsRNA. Upon pupation, both male and female control individuals were phenotypically identical to wild-type beetle pupae (Fig. 1b, far left and right). In contrast, *OtdsxC* dsRNA-injected pupae exhibited intersex phenotypes, as evidenced by the greatly reduced head horns in males (Fig. 1b, centre left) and the presence of ectopic head horns in females (Fig. 1b, centre right). Although head horn size was similar in both male and female *OtdsxC* RNAi individuals, the overall difference between control and *dsx*RNAi individuals was notably larger for males than females (Fig. 1c). Taken together, our results faithfully replicated all previously reported *OtdsxC* knockdown phenotypes[14].

**Differential expression analyses.** We first sought to characterize the extent and distribution of sex-biased gene expression across tissue types in control individuals. Our comparison of control females with males identified 2,720 genes with significantly female-biased and 1,565 genes with significantly male-biased expression ($P_{adj} < 0.05$; Supplementary Fig. 1). The majority of genes exhibiting sex-biased expression were found in genital tissue (85% and 67% of genes with female- and male-biased expression, respectively), with the next largest proportion of sex-biased gene expression occurring uniquely in head horn tissue (8% and 26% of genes with female- and male-biased expression, respectively). Conversely, brain tissues exhibited a paucity of sex-biased expression, with only three and seven genes exhibiting female and male-biased expression, respectively. Importantly, relatively few genes with sex-biased gene expression were shared over 2, 3 or 4 tissues, making up only 5% each of the total number of genes with female- and male-biased gene expression.

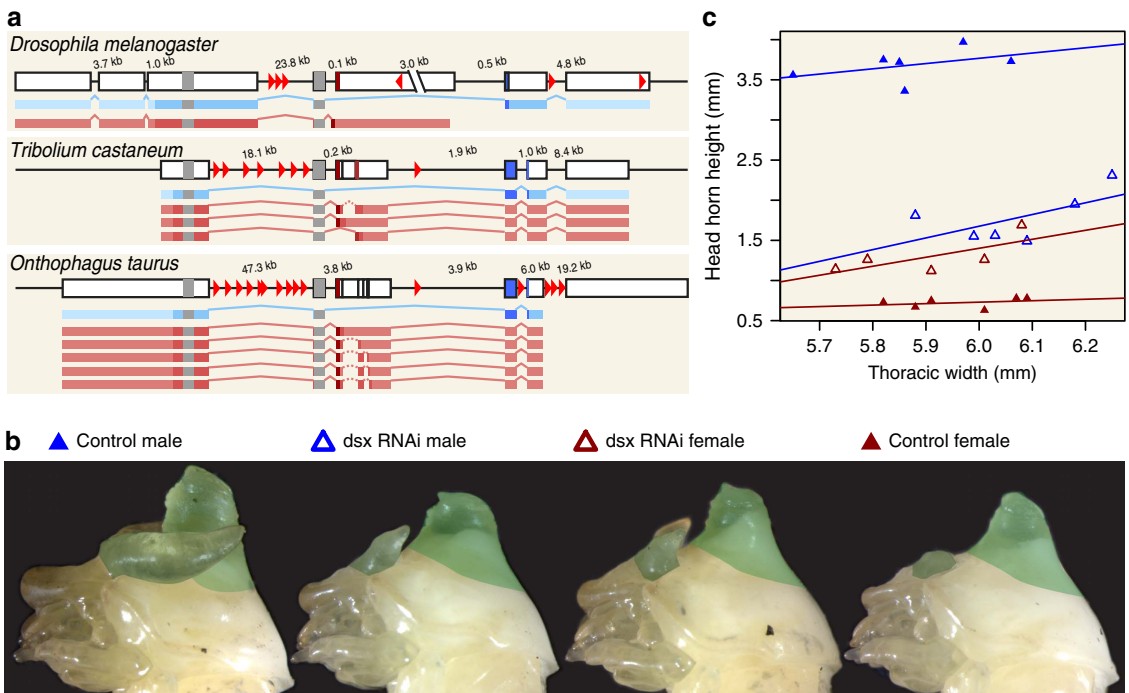

**Figure 1 | *dsx* genomic structure and RNAi phenotypes.** (**a**) Genomic structure and isoforms of *dsx* in the fruit fly *Drosophila melanogaster* (Diptera; top), red flour beetle *Tribolium castaneum* (Coleoptera: Tenebrionidae; middle) and bull-headed dung beetle *Onthophagus taurus* (Coleoptera: Scarabeidae; bottom). Each gene is composed by several exons (black-border boxes) connected by intronic sequences (black line). Male (blue bars) and female (pink bars) isoforms are shown below each gene; within each isoform, a darker hue indicates the coding sequence, the two grey regions represent parts of the OD1 (leftmost) and OD2 (rightmost) domains that are common across all isoforms, and the darkest regions represent sex-specific regions of the OD2 domain. Solid lines in each isoform represent canonical splice junctions; dashed lines indicate non-canonical splicing within a female-specific exon (in beetles). Red arrowheads show positions of putative *dsx*-binding sites. Exons are drawn to scale; introns are not to scale, but their sizes are indicated above each intron. (**b**) Representative individuals showing the pupal phenotype after sham control dsRNA injection (leftmost: male; rightmost: female) and *OtdsxC* dsRNA injection (centre left: male; centre right: female). Head horn and thoracic horn tissues dissected for RNAseq false coloured in green. (**c**) Head horn size of the pupae used for dissection, plotted against pupal weight. *OtdsxC* dsRNA injection (open triangles) reduces head horn size in males (blue triangles) but induces head horn growth in females (red triangles) compared to sham controls (filled triangles).

We then addressed whether Dsx has a role in generating genome-wide sex-biased gene expression. To do so, we first focused on genes we had identified in the previous comparison as significantly sex-biased in control animals, and found that their corresponding levels of expression in *dsx*RNAi animals were generally more similar between the sexes. Put another way, the average degree of sex-bias normally shown by these genes in control individuals was reduced in *dsx*RNAi individuals (Fig. 2a), and manifested as fewer significantly sex-biased genes across tissues (Fig. 2b), resulting in an approximately threefold reduction in the number of significantly sex-biased genes, from 4,285 (control) to 1,611 (*dsx*RNAi; $P_{adj} < 0.05$).

Next, we asked whether Dsx mediates genome-wide sex-biased gene expression via inhibition or activation of gene expression across the sexes. To do so we compared the transcriptomes of *dsx*RNAi males with those of control males, as well as those of *dsx*RNAi females to those of control females (Supplementary Data 1). We found that in *dsx*RNAi males, several genes are upregulated relative to control males in all four tissues (Fig. 2c), suggesting that Dsx normally directly or indirectly suppresses gene expression in males. Likewise, we found that in *dsx*RNAi males, several genes are downregulated relative to control males, suggesting that Dsx also normally directly or indirectly activates gene expression in males. In partial contrast, several genes were upregulated in *dsx*RNAi females relative to control females, but solely in head horn tissue. Furthermore, in *dsx*RNAi females, there was limited downregulation of genes in any tissue relative to control females, suggesting that Dsx has a limited role in

activating gene expression in females. Combined, these data suggest that Dsx mediates downstream gene expression by both activation and repression, and that the strength and direction of its influence are highly context-dependent, showing asymmetric effects for different tissues and sexes.

Next, we focused our analysis on the gene repertoire identified in control individuals as exhibiting sex-biased expression to further assess the dependency of sex-biased expression on Dsx for each tissue by sex combination. We detected two major patterns. First, in two tissues (genitalia and thoracic horns), genes whose expression exhibited male bias in control individuals generally exhibited reduced expression in *dsx*RNAi males, approximating the expression levels of the same genes in control females, and genes whose expression normally exhibits female bias in control individuals generally exhibited increased expression in *dsx*RNAi males, now approximating the expression levels of the same genes in control females (Wilcoxon test, $P_{adj} < 0.05$, Fig. 3a upper left; Supplementary Fig. 2). The same patterns were also observed in male brains, although these trends were not significant (Supplementary Fig. 2). The reverse pattern in females, however, was not observed across these three tissues; the degree of male-biased and female-biased gene expression did not increase or decrease, respectively, in *dsx*RNAi knockdown females (Fig. 3a, upper right; Supplementary Fig. 2).

The second pattern was observed in head horns only, and involved changes in gene expression for both sexes. In this tissue, *dsx*RNAi males exhibited a reduced degree of male-biased gene expression and an elevated degree of female-biased gene

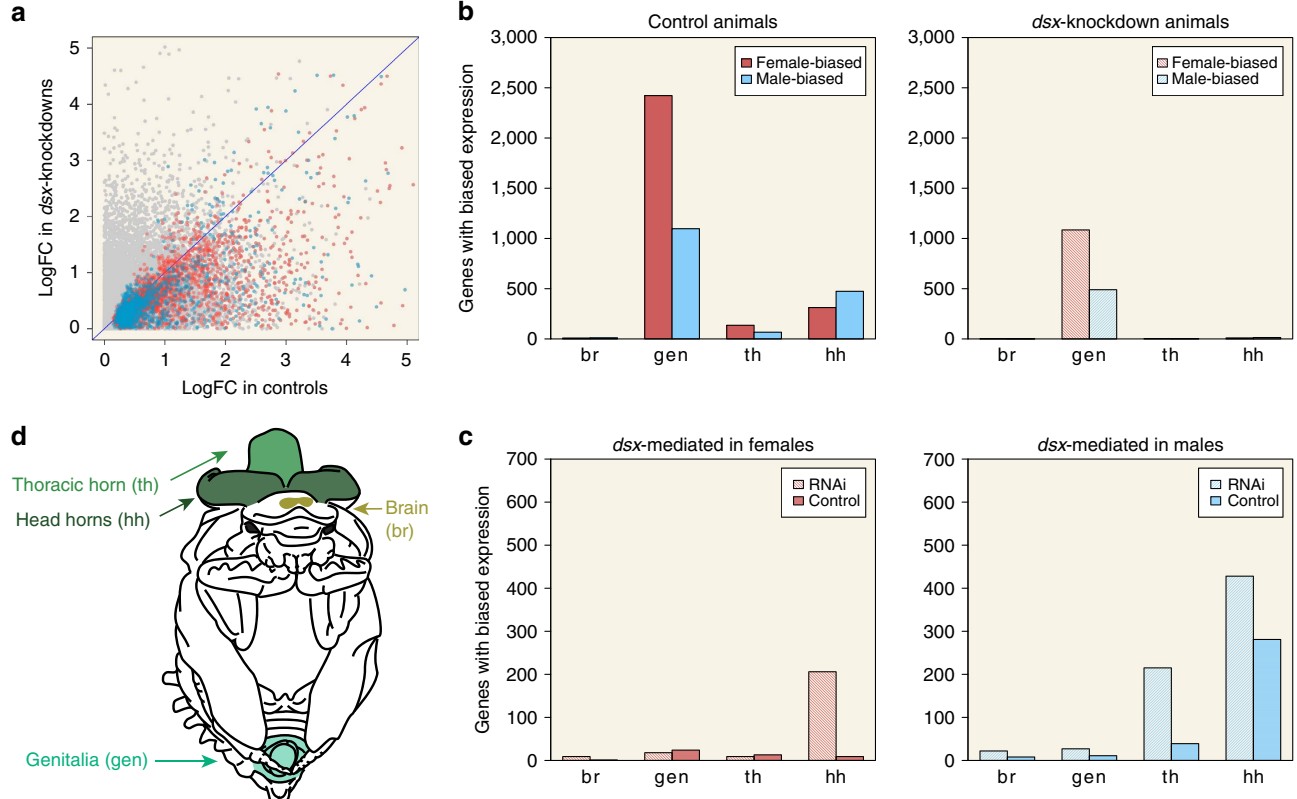

**Figure 2 | *dsx* RNAi reduces sex-biased gene expression across horned beetle tissues.** (**a**) Degree of significantly ($P_{adj} < 0.05$) female- (pink) or male-biased (blue) gene expression in control animals measured as log-transformed fold change in FPKM (x axis) plotted against the corresponding level of expression in *dsx*RNAi animals (y axis). Genes that did not have significantly sex-biased expression are plotted in grey. Blue line reflects equally sex-biased expression in control and *dsx*RNAi individuals. (**b**) *dsx*RNAi reduces the number of *O. taurus* genes exhibiting significant sex-bias in control individuals. The number of genes with significantly ($P_{adj} < 0.05$) female-biased (solid pink) or male-biased (solid blue) expression in control animals (total = 4,285) was reduced ~2.7-fold when assessed in *dsx*RNAi animals (pink and blue lines; total = 1,596). (**c**) *dsx*RNAi females (pink lines) overexpress more genes relative to control females (solid pink) in head horns only, indicating that, in females, Dsx directly or indirectly suppresses the expression of downstream genes in that specific tissue. *Dsx*RNAi males (blue lines) also overexpress as many or more genes relative to control males (solid blue), but do so across tissues, indicating that, in males, Dsx acts to directly or indirectly suppress the expression of downstream targets across a broader range of target tissues. In addition, in head and thoracic horns, control (solid blue) males overexpress many genes relative to *dsx*RNAi males (blue lines), suggesting that, in males but not females, Dsx also directly or indirectly activates the expression of many downstream targets. (**d**) Tissues analysed in this study were derived from 1st day pupae, and included brains (gold), genitalia (teal), thoracic horn (light green) and head horns (dark green).

expression (Fig. 3b; lower left), whereas *dsx*RNAi females solely exhibited an elevated degree of male-biased gene expression (Fig. 3b; lower right). Together, these data suggest that Dsx directly or indirectly modulates sex-biased gene expression across all four tissues assessed in this study (brains, genitalia, thoracic horns and head horns) in males, but only in a single tissue in females (head horns) by specifically suppressing genes with male-biased expression (Fig. 3c).

**Sex-biased genes at a functional level.** Our enrichment analysis was performed to determine what functional processes are overrepresented among genes with sex-biased or Dsx-mediated expression (Supplementary Data 2). We found that genes with sex-biased expression in head horn tissue are enriched for functional processes related to immunity and cellular taxis. Although immunity might seem unrelated to horn development, it is worth noting that this category included genes involved with morphogenesis (*dachs*; *lola*), cell growth and death (*ago*; *syx5*; *traf6*), the dopamine biosynthetic pathway (*tan*) and dorsoventral patterning of the early embryo (*dl*; *tl*), processes known or suspected to contribute to horn development[13,15,16]. In addition, we found that sex-biased genes expressed in thoracic horns were

enriched for functional processes related to WNT signalling, extracellular matrix construction, receptor binding and axon development.

We also evaluated the enrichment of functional processes in genes whose expression was Dsx-mediated and found that Dsx-mediated genes in male head horns were enriched for MAP kinases, threonine kinases and functions involved with receptor signalling, whereas genes that were Dsx-mediated in male thoracic horns were enriched in functional processes related to chitin and cuticle construction, sensory perception and sugar metabolism. For Dsx-mediated genes in female head horns, we found that they were enriched for the functional categories of extracellular matrix construction, and ion channels, whereas those Dsx-mediated in female thoracic horns were enriched mainly in functional categories related to metabolic processes.

Lastly, we compared our functional categories with those identified by a previous study on fruit flies wherein the authors performed an enrichment analysis to determine what gene ontology categories were overrepresented among 3,717 putative Dsx targets[9] (Supplementary Data 3). Fly putative Dsx targets and horned beetle sex-biased genes were both enriched for immune system processes, receptor binding, axogenesis and WNT signalling functional categories; fly putative Dsx targets and

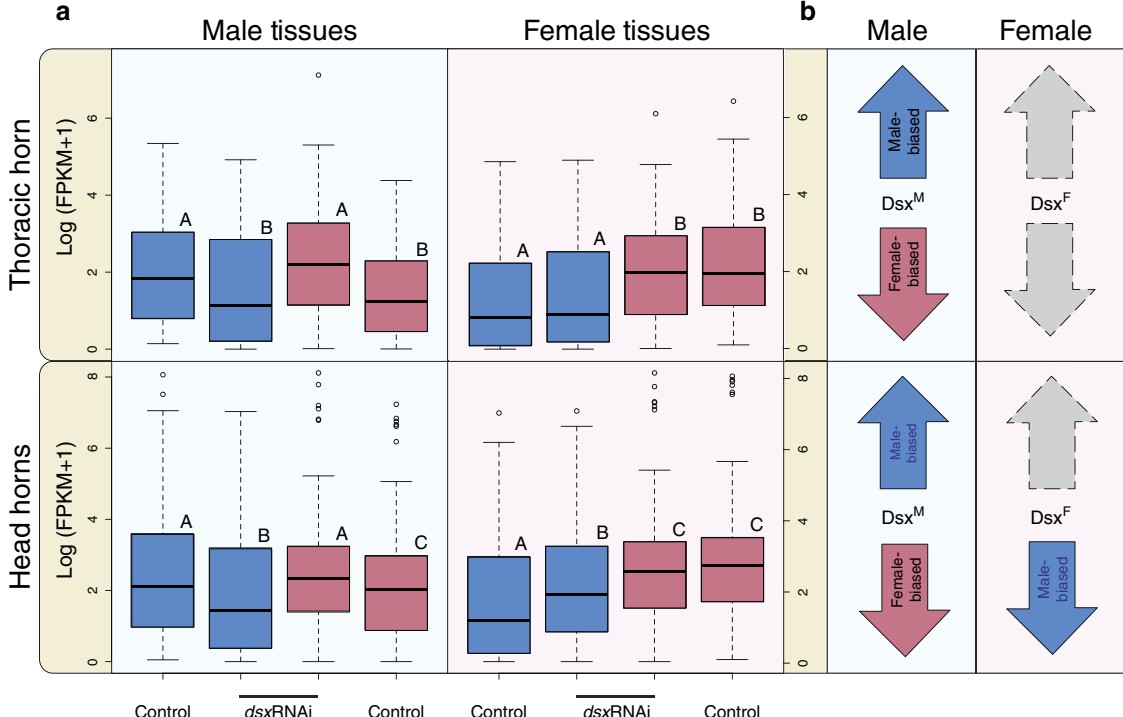

**Figure 3 | *dsx*RNAi affects male- and female-biased gene expression in a sex-specific and tissue-specific manner.** Dark blue and pink represent genes with male- and female-biased expression, respectively, as determined by comparisons between control males and control females. Background colours indicate whether those genes were assessed in male (light blue) or female (light pink) tissues. Letters indicate significant differences between levels of gene expression (Wilcoxon test and Benjamini–Hochberg correction for multiple comparisons). (**a**) In all male tissues assessed (thoracic and head horns pictured here, brain and genitalia in Supplementary Fig. 2), *dsx*RNAi decreased levels of male-biased gene expression (dark blue) and increased levels of female-biased gene expression (dark pink). In most female tissues (thoracic horn pictured here, brain and genitalia in Supplementary Fig. 2), *dsx*RNAi neither decreased levels of female-biased gene expression nor increased levels of male-biased gene expression. However, uniquely in female head horns (pictured here), *dsx*RNAi resulted in increased levels of male-biased gene expression. Lines within boxes are median values of gene expression and whiskers extend to the most extreme data points that are not outliers. Outliers—data points greater than (the upper quartile + 1.5 × the interquartile range) or less than (the lower quartile − 1.5 × interquartile range)—are represented as circles. (**b**) A summary of the differential actions of male and female Dsx isoforms among tissues; blue arrows indicate the effect on genes with male-biased expression, pink arrows indicate the effect on genes with female-biased expression and grey arrows indicate the absence of an effect. These results show that the male Dsx isoform modulates sex-biased gene expression across several male tissues, but that female Dsx isoforms typically only modulate male-biased gene expression in a single female tissue: head horns.

horned beetle Dsx-mediated genes were both enriched for receptor binding and regulation of morphogenesis functional categories.

**Binding-site analysis**. To define the target repertoires of Dsx across sexes and tissues in *O. taurus,* we then sought to identify genes likely targeted directly by Dsx and distinguish them from genes whose expression changes occur as an indirect consequence of Dsx modulation. We defined putative direct Dsx targets as genes (i) whose expression exhibited differential expression between control and *dsx* mRNA knockdown animals and (ii) which possessed five or more significant predicted Dsx-binding sites (Supplementary Data 4). We then compared the composition of these putative Dsx target repertoires across sexes and tissues. We found that Dsx targeted largely non-overlapping gene repertoires in homologous tissues of males and females, irrespective of whether Dsx was suppressing or activating the expression of those genes (Supplementary Data 4). For example, although Dsx directly targeted 36 and 31 genes in female and male genitalia, respectively, none of these genes were the same. Genes putatively targeted by Dsx in brains and thoracic horns also exhibited this pattern. The only exception to this pattern occurred in head horns where a surprisingly large number of the

same genes were affected in opposing directions in male and females, accounting for 8% of all head horn genes putatively targeted by Dsx. In contrast, genes targeted in both male and female head horns that were modulated in the same direction constituted only 0.3% of Dsx-targeted genes in head horns. This finding suggests that although Dsx acts independently between the sexes in most *O. taurus* tissues, it can also work as a switch to generate alternate sexual phenotypes in head horn tissues.

Lastly, we sought to determine which, if any, of these putative targets of Dsx were unique or shared across tissues within a sex. Among tissues (but within sexes), the majority of Dsx targets were uniquely expressed in head horn tissue (75% and 66% of all female and male targets, respectively; Supplementary Fig. 3). The remaining Dsx targets were concentrated in single tissues in both sexes (for example, 16% of targets of the male isoform are expressed exclusively in thoracic horn tissue). The only exception to this pattern was that male head and thoracic horn tissue shared a sizeable number of putative Dsx targets (11% of all male targets), whereas the homologous tissues in females shared none (Supplementary Fig. 3). Collectively, the data from the binding-site analysis support the notion that Dsx directly influences the expression of its target genes in a largely tissue- and sex-specific manner, with the exceptions being that (i) a considerable number of putative targets expressed within males are shared between

head and thoracic horns, and (ii) a considerable number of putative targets expressed within head horns are both shared and oppositely modulated by Dsx in males and females.

As a final comparison we contrasted Dsx targets identified in this study to those identified in a study of the fruit fly *Drosophila melanogaster*[9]. Using BLAST searches we found 603 *O. taurus* orthologs corresponding to the Dsx targets identified in the fly study. Of these, 444 were also found by our binding-site analysis, such that ∼74% of the Dsx targets identified in the fly study were also recovered in our study (Supplementary Data 5). Interestingly, both the fly study and our study identified *dsx* as a target of itself.

**Candidate pathways and genes for future functional studies.** Our analyses identified several candidate pathways and genes as putative Dsx targets, which had independently been implicated in the regulation of sex-specific development by previous studies, such as ecdysteroid and *hedgehog* signalling pathways[13,17]. We also identified several other prominent and well-studied pathways as putative *O. taurus* Dsx targets whose roles in *Onthophagus* development remain to be characterized, such as the JNK signalling pathway[18] or circadian regulators[19,20]. More generally, and consistent with several recent studies on other insects, *O. taurus* Dsx appears to target a wide diversity of genes, including both upstream regulatory genes as well as more terminal effector genes[21–23]. Our study similarly found that Dsx putatively targets genes that are both at the top (for example, *homothorax*, *distal-less* and *pumilio*) and bottom (for example, chitinases, cathepsin and myosins) of developmental hierarchies. Collectively, these results provide valuable substrate for future functional analyses.

**Discussion**
Understanding how developmental-genetic mechanisms enable, bias or constrain the often rapid evolution of sex-specific differentiation in insects and beyond necessitates an understanding of the context-dependency of Dsx target gene repertoires. In this study we used the sexually dimorphic bull-headed dung beetle *O. taurus* to investigate Dsx's target repertoires as a function of sex and tissue type. Our approach employed the functional knockdown of *dsx* using RNAi in concert with genome-wide analyses of gene expression. We found that (i) Dsx has a central role in generating sex-biased gene expression by both activation and suppression of gene expression; (ii) whereas Dsx modulates sex-biased gene expression across several male tissues, its role in homologous female tissues is mostly restricted to head horn tissue; and (iii) the target repertoire of Dsx in head horns is unique in that it is both exceptionally large relative to the other tissues assessed, and because some of these targets are shared between males and females—albeit modulated in opposite directions—whereas targets among other tissues are largely sex-specific. Together, our results suggest that the target repertoires of Dsx, although asymmetric in size between sexes and among tissues, can evolve rapidly to accommodate novel, sexually dimorphic traits, whether governing gene expression through sex-specific or switch-like mechanisms.

Consistent with previous studies in *O. taurus* that assessed sex-biased gene expression across a subset of the same tissue types used in this study[24–27], our control comparisons revealed widespread female- and male-biased gene expression, most of which was also tissue-specific. Our finding that genes with female- or male-biased expression in control animals became more similar in expression when assessed in *dsx*RNAi knockdown animals is also consistent with the existing literature implicating Dsx as a master regulator of sexual

differentiation[1,28,29]. Further, we found that, across tissues, Dsx generates genome-wide sex-biased gene expression via both the activation and suppression of downstream genes. Although Dsx's activating and suppressing nature has been described for several candidate genes[1], our study reveals that Dsx may be exerting this role broadly across the genome. Two patterns are perhaps most surprising; first, both female and male Dsx isoforms actively repress gene expression in head horns. Although previous work has determined that female hornlessness relies at least in part on the repression of gene expression[14,27], our results here indicate that the explosive growth of male head horns may similarly require the repression of several genes.

Second, although we detected nearly twice as many genes with female-biased as opposed to male-biased expression among control animals, many fewer genes were in fact modulated by Dsx in females than in males across tissues. Given the wealth of genes with female-biased expression, these data suggest that Dsx-independent mechanisms must exist to account for non-Dsx modulated sex-biased gene expression, a finding that is not entirely unexpected. One potential Dsx-independent mechanism for producing female-specific phenotypes is the gene *hermaphrodite* (*her*), a zinc finger protein that can likely regulate female-biased gene expression directly. In *D. melanogaster*, the *her* pathway sometimes interacts with the female-specific Dsx protein, but can also function independently of Dsx to control downstream target genes in cells that give rise to female-specific traits[30]. In our study, expression levels of *her* did not change significantly in response to *dsx* knockdown (Supplementary Data 1), which would be consistent with *her* acting independently of Dsx.

Next we investigated how Dsx modulates, in each sex, genes that have female-biased or male-biased expression (as assessed in control animals). We found that in males, Dsx modulates male-biased and female-biased expression in all four male tissues assessed, but in females it solely modulates male-biased expression, and only in a single tissue: head horns. In this female tissue, the female Dsx isoforms suppress many of the same genes that are activated by the male Dsx isoform in males. This type of antagonistic gene regulation between the sexes, observed uniquely in head horn tissue, may provide a mechanism to amplify or refine preexisting somatic sex-biased growth in a sexually selected trait (of the traits assessed in this study, only head horns are sexually selected[31]). Although males may experience sexual selection to grow larger horns, females may experience natural selection to suppress horns; indeed, horns inhibit manoeuverability in tunnels, which is essential for females who assume the burden of constructing tunnels and broodballs[32]. Genes underlying horn growth might then be sexually antagonistic[33]; the ability of the female Dsx isoforms to repress the same genes that are promoted by the male Dsx isoform may alleviate the consequences of such sexual antagonism, resulting in the rapid evolution of novel Dsx targets in female head horns relative to other female traits that are not sexually selected, a pattern we predict might also be found in other sexually selected traits such as the eye stalks of diopsid flies or the mandibles of stag beetles.

The observation that the contributions of male and female Dsx isoforms to sex-biased gene expression are asymmetric among several tissues (brain, genitalia and thoracic horn) is not altogether unexpected because the alternate splicing of *dsx*, and therefore the existence of sex-specific isoforms and their discrete roles in sexual differentiation, is itself a derived phenomenon[34]. Although basal insect lineages possess alternately spliced isoforms of Dsx, other arthropods outside the Hexapoda do not, nor do more distantly related Ecdysozoa[3]. In the crustacean *Daphnia*, for instance, no alternate male and female isoforms are produced, and although Dsx is essential for male development it is not necessary for female differentiation[35]. Similarly, in the nematode

*Caenorhabditis elegans* (which also lacks sex-based splicing), DMRT expression is necessary for male differentiation, but non-essential for the development of hermaphrodites[36]. The significance of *Dmrt* genes for promoting male-specific development even has parallels in vertebrates; in mouse, DMRT1 is responsible for maintaining male and suppressing female gonadal identity[37]. Finally, correlative studies demonstrate a relationship between DMRT1 and male-specific development across several taxa including flatworms, molluscs, amphibians, turtles and crocodiles[38]. Thus, if the sex-specific splicing of *dsx* in insects derives from a non-spliced ancestral condition where Dsx had a role in promoting male development only, it is possible that the targets and functions of the female Dsx isoforms are relatively novel (Fig. 4) and less numerous than those of the male isoform.

Beyond understanding how genes are generally modulated by Dsx, identifying the repertoire of genes directly targeted by Dsx is critical for understanding how sexual differentiation is achieved during development and how sexual dimorphisms may be modified during evolution[8,39]. We used a genome-wide bioinformatic approach to determine which of the genes whose expression changed significantly after our *dsx* knockdown also harboured predicted Dsx-binding sites (hereafter, 'putative Dsx targets'). Our findings in *Onthophagus* agree with those of two functional studies in *Drosophila* involving Dsx-binding sites in the enhancer regions of genes *yellow*[40] and *desatF*[41], suggesting that our approach was successful in identifying likely and biologically relevant targets of Dsx at a genome-wide level. Generally, our binding-site analysis revealed that most genes whose expression is modulated by Dsx also contain multiple Dsx-binding sites, that is, qualify as putative Dsx targets. More specifically, our analysis shed light on at least two critical aspects of Dsx's role in generating sexual dimorphism. First, Dsx targets are often tissue-specific and vary widely among body regions. Although previous studies have emphasized the importance of changes in Dsx's spatial expression in evolution[42], here we show that the target repertoire of Dsx may be similarly evolutionarily labile across tissues and may diverge rapidly to accommodate novel traits and patterns of trait expression. Second, most of these putative targets were regulated only in males or females, but in head horns, a sizeable number were regulated in both sexes but in opposite directions. These findings highlight two distinct avenues by which Dsx may directly mediate sexually dimorphic development in homologous structures in males and females. In tissues like thoracic horns and genitalia, Dsx appears to directly target essentially non-overlapping gene sets in males and females such that male and female patterns of gene activation and repression in those tissues are independent of each other. In contrast, in head horns, Dsx appears to act like a switch mechanism, activating the same genes in males that are being repressed in females. These results match findings from several other studies demonstrating that male and female Dsx isoforms modulate target genes in both opposing[39,43,44] as well as sex-independent directions[30,41].

Taken together, our assessment of Dsx targets suggests that the size and nature of Dsx's target repertoires varies considerably between tissues and among sexes. This remarkable ability of Dsx to differentially regulate diverse suites of genes between tissues of the same sex may be due to the differential presence of cofactors among tissues that interact with Dsx or with *cis*-regulatory elements that direct transcription in a tissue-specific manner[45]. In the same vein, Dsx may be able to differentially direct expression between sexes in homologous tissues owing to the distribution of sex-specific cofactors, or alternatively, the differing properties of its male and female isoforms that possess male-specific and female-specific oligomerization domains (that is, the domains that interact with cofactors[10]). For example, in *Drosophila*, the

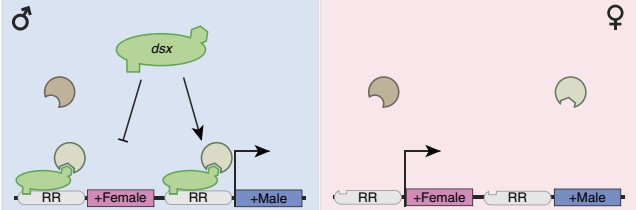

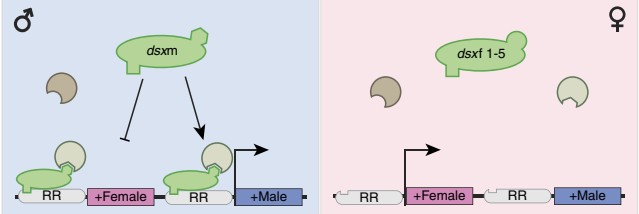

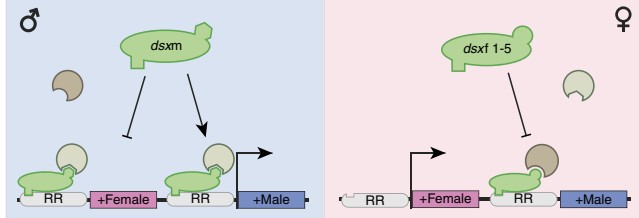

**Figure 4 | Dsx asymmetrically directs sex- and tissue-specific gene expression in most *O. taurus* tissues, but acts as a genetic switch in head horns, a recently evolved and sexually selected trait.** (**a**) In non-insect arthropods retaining the putative ancestral condition, Dsx possesses only a single isoform, and contributes mostly to the masculinisation of male tissues (left, blue background) by promoting masculinising genes ( + male boxes) and suppressing feminising genes ( + female boxes), but has no role in female-specific differentiation (right, pink background). (**b**) In insects, Dsx is alternately spliced into male- and female-specific isoforms. In *O. taurus*, the species addressed in this study, Dsx acts as a sex-specific modulator in several tissues (brain, genitalia and thoracic horns), such that the target gene repertoires of female and male isoforms are largely independent. Further, although females express feminising genes across these tissues, the female Dsx isoforms do not have a large role in mediating such expression. The ability of male and female isoforms to target different gene repertoires in spite of identical DNA-binding domains is the result of either differential presence of cofactors between sexes (not pictured) or the different interactions with cofactors that arise from the different C-terminal oligomerization domains of the male and female isoforms (pictured). In addition, although the target gene repertoires of female and male Dsx isoforms are largely independent in these tissues, the male isoform can also suppress the expression of feminising genes in males, even when the same genes are not regulated by the female Dsx isoform in females. (**c**) Specifically in *O. taurus* head horn tissue, which yields a sexually dimorphic trait subject to strong sexual selection, Dsx also acts as a sexually differentiating switch. Uniquely, male and female isoforms modulate the same masculinising genes, albeit in opposite directions. Whereas in male head horn tissue the male Dsx isoform activates masculinising genes, in the female head horns, the female Dsx isoforms suppress the same masculinising genes. RR: regulatory region.

DNA-binding domain of Dsx is shared by the male and female isoforms, but the C-terminal amino acids are encoded by female- and male-specific exons. These differences manifest as different cofactor interactions with Dsx between the sexes in homologous tissues. For instance, the intersex protein is produced in both

males and females, but interacts only with Dsx$^F$ and not Dsx$^M$ [46,47]. A similar phenomenon likely occurs in *O. taurus*, where—as in *Drosophila*—sex-specific isoforms possess the same functional DNA-binding and oligomerization domains, but the female *dsx* transcripts (*Otdsxf* 1–5) possess translation termination codons that are absent from the male *dsx* transcripts[14] (Fig. 1a), likely rendering them functionally divergent.

Our analysis also revealed diverse candidate genes and pathways that possibly underlie sexually dimorphic development (Supplementary Data 4), specifically of horn tissues. Interestingly, we found that Dsx may be regulating itself in male head and thoracic horns by activating its own expression. In contrast, Dsx does not activate its own expression in female head or thoracic horns, not because Dsx is not present to begin with (its presence in females has been previously confirmed by real-time PCR[14]), rather, the two isoforms must be differentially interacting with cofactors that influence their own expression.

Of the 54 genes that are shared between but differentially modulated by male and female Dsx isoforms in head horn tissue, 53 are activated by the male Dsx isoform (and suppressed by the female Dsx isoforms) and only 1 is activated by the female Dsx isoforms (and suppressed by the male Dsx isoform). The gene activated by the female *dsx* isoforms is the transcription factor cycle (also known as ARNTL and BMAL-1[19]), a well-known circadian rhythm regulator[20]. This finding is intriguing, given that other circadian rhythm genes have been identified as having sex-biased patterns in *O. taurus* head horns using independent methods[27].

Additional target genes that were downregulated by Dsx in male head horns were involved with Wnt and Hedgehog signalling (for example, *apc-like*, *frizzled 2, sulfateless* and *smoothened*), two major and often interacting pathways with prominent roles across metazoan development[48]. This observation matches predictions that can be derived from a recent study on the role of *hedgehog* signalling in *Onthophagus* development, which found that downregulation of *smo* induced large horns in nutrient limited males which normally do not develop them[17]. In contrast, under high nutrition (as in the males analysed here) horn growth is promoted by the male Dsx isoform, likely via the direct inhibition of *smo*. In addition, in male head horn tissue, the male Dsx isoform inhibited the expression of genes involved with ecdysteroid signalling (for example, *shade, phantom, centaurin gamma 1A, eclosion hormone* and *ecdysone-inducible gene E1*), which has a central role in moulting and metamorphosis[49] and has recently been implicated it the regulation of sex-specific development in horned beetles[13,50].

In summary, the study presented here aimed to characterize the sex- and tissue-specific genetic target repertoire of Dsx in an onthophagine beetle, *O. taurus*. Our results suggest that Dsx may be uniquely positioned to rapidly accommodate the sex-biased development of evolutionarily novelties, as evidenced by the exceptionally large and unique sets of genetic targets mediated by Dsx in both male and female head horns. These findings provide a useful baseline for investigating how Dsx repertoires may change along both developmental and evolutionary axes: for example, as is true for many *Onthophagus* species, *O. taurus* males subject to low nutrition develop into hornless minor males resembling females, and it will be interesting to determine whether, under low nutritional conditions, the male Dsx isoform converges on the same targets as the female isoforms, or whether both produce similar developmental outcomes via different genetic targets. Similarly, patterns of sex-biased horn expression have diverged markedly among the over 2,000 extant onthophagine species[51], and future comparative studies will now be in a position to determine the degree to which changes in the interactions between Dsx and its targets have facilitated this

dramatic radiation of secondary sexual traits. More generally, the results from our study raise the possibility that comparative analyses of Dsx-target interactions in these and other species may provide key insights into the developmental-genetic mechanisms, including their biases and constraints, that shape the development and evolution of sexual dimorphisms in insects and beyond.

## Methods

**Animal collection and husbandry.** *O. taurus* beetles were field collected near Bloomington, IN and Chapel Hill, NC, kept in a sand/soil mixture at a 16:8 h light:dark cycle at 25 °C, and fed homogenized cow manure twice a week. Animals used for experiments were generated by breeding five females and three males in a moist sand/soil mixture container with *ad libitum* food. After 8 days, broodballs produced by females were collected and larvae were transferred to 12-well plates containing artificial broodballs. Artificial broodballs were created by filling wells (1.5 cm deep and 1.8 cm diameter) with cow manure that had been previously drained of water to mimic the consistency of broodballs naturally produced by adult beetles[52].

**dsRNA synthesis and injection.** Synthesis and injection of the *OtdsxC* dsRNA construct to knock down both male and female mRNA *dsx* isoforms were performed as previously described[14,53]. In brief, a pSC-A vector containing the *OtdsxC* construct was used as template in PCR reactions using M13 forward and reverse primers. Primers sequences (from 5′ to 3′) for *OtdsxC* were: TGATTCC CCAATCGAAAA and TTTGGCCAATATTGTTATTCC. The resulting PCR product was used as a template for *in vitro* transcription of forward and reverse RNA strands using MEGAscript T7 and T3 Kits (Life Technologies). After DNase I treatment, single-stranded RNA was precipitated by lithium chloride, incubated at − 20 °C for 1 h, spun at 4 °C for 20 min, washed with 80% ethanol and resuspended in water. After quantification, forward and reverse strand RNAs were mixed at a 1:1 ratio by weight, and annealed by heating to 80 °C and then cooling slowly over 5 h to 35 °C. The concentration of the annealed dsRNA was measured, confirmed by gel electrophoresis and stored at − 80 °C until injection. Injections into *O. taurus* larvae were carried out as described previously[14,53], with doses of 0.5 μg of dsRNA diluted in injection buffer (5 mM KCl, 1 mM KPO4 pH 6.9) to a total of 3 μl and injected into larvae during the first 5 days of the final instar. Sham control injections were made exactly as described above, except that larvae were injected with 1 μg of control dsRNA generated from a 167-bp PCR product derived from a pBluescript SK vector. Transcription reactions, DNase I treatment, transcript annealing and injections were performed as described above. The efficiencies of the dsRNA injections were confirmed via high-throughput RNA sequencing, and are provided in Supplementary Fig. 4

**Sample preparation and RNA extraction.** Individuals from both *OtdsxC* and sham control dsRNA injection treatments were phenotyped, imaged and weighed within the first day after pupation. Six male and six female pupae weighing over 125 mg from each group were used for tissue dissection and total RNA extraction. Each pupa was rinsed with RNAse-free distilled water, submerged in 0.05% Triton-X in phosphate buffered saline and dissected. We sampled tissue from four distinct regions (Fig. 2d): the supraesophageal ganglion (hereafter, brain tissue), the abdominal epidermis of the 4–5 posterior most segments (which include the external male copulatory organ), the dorsal region of segment T1 (which includes thoracic horns in all pupae), and the posterodorsal head epithelium (where head horns are located in control males and *OtdsxC* knock-downs). All tissue samples were immediately placed in ice-cold Trizol (15596018, Thermo Fisher Scientific), and stored at − 80 °C until further processing. After thawing to 4 °C, tissues in Trizol were homogenized with disposable polypropylene RNase-free pestles, and total RNA was extracted using a standard phenol–chloroform protocol followed by RNeasy Mini (74104, Qiagen) spin column purification including on-column DNAse I digestion.

**Library construction and high-throughput sequencing.** Total RNA was quality checked using an Agilent 2200 TapeStation system with an RNA ScreenTape Assay and quantified with a Quant-iT RiboGreen Assay Kit (Thermo Fisher). A total of 96 (two treatments × two sexes × four tissues × six biological replicates) RNA Stranded RNA sequencing libraries were constructed using the TruSeq Stranded mRNA Sample Preparation Kit (Illumina, San Diego, CA) according to manu-facturer's instructions. Libraries were quantified using a Quant-iT DNA Assay Kit (Thermo Fisher), pooled in equal molar amounts and sequenced as single-end reads using a 75-cycles High kit on the NextSeq500 platform (Illumina, San Diego, CA). Resulting reads were cleaned using Trimmomatic version 0.32 (ref. 54) to remove adapter sequences and perform quality trimming. Trimmomatic was run with the following parameters, '2:20:5 LEADING:3 TRAILING:3 SLIDINGWINDOW:4:15 MINLEN:18'. The trimmed reads where then reversed complemented using FASTX-Toolkit version 0.0.13.2 (http://hannonlab.cshl.edu/fastx_toolkit/download.html), and mapped against the *O. taurus* genome v0.5.3 gene models[55] using TopHat2 version 2.1.0 (ref. 56) with the parameters '—b2-

very-sensitive—read-edit-dist'. Read counting was done for each gene using the htseq-count command from the HTSeq package version 0.6.1p1 (ref. 57). A table containing counts normalized per kilobase of feature length per million mapped fragments (FPKM values) was generated using the R package DESeq2; these values were used to visually assess correlations among all replicates and statistically test correlations between biological replicates within sample groups using Pearson's correlation coefficients (Supplementary Fig. 5; Supplementary Data 6).

**Differential expression analyses.** To explore expression differences between *O. taurus* sexes, tissues and treatment groups (control and *OtdsxC* dsRNA-injected), we tested for differential gene expression across several comparisons using the R package DESeq2 that, in a pairwise fashion, employs negative binomial modelling and adjusts for multiple testing using the Benjamini–Hochberg method[58].

First, to determine whether *dsx* has a role in generating sex-biased gene expression, we identified significant expression differences ($P_{adj} < 0.05$, a $P$ value adjusted with a false-discovery rate correction using the Benjamini–Hochberg method[59]) between control females and males (CF-CM) and between *OtdsxC* dsRNA-injected females and dsRNA-injected males (DF-DM) across four tissues (brain, genitalia, thoracic horn and head horns). We compared the degree of biased expression, as measured by FPKM (Fragments Per Kilobase per Million reads), generated by each contrast. Results were plotted, for each gene, as the log-transformed fold change (logFC) difference in gene expression for a given contrast. In addition to comparing how the levels of bias in each gene's expression changed across control (CF-CM) and dsRNA-injected (DF-DM) animals, we determined how the number of significantly sex-biased genes ($P_{adj} < 0.05$), as determined in control animals, changed when the same genes were assessed in *OtdsxC* dsRNA-injected animals.

Second, to determine the set of genes that are mediated (directly or indirectly) by Dsx, irrespective of whether the gene activates or suppresses expression, we identified significant expression differences ($P_{adj} < 0.05$) between *OtdsxC* dsRNA-injected females and control females (DF-CF) and between *OtdsxC* dsRNA-injected males and control males (DM-CM) across four tissues (brain, genitalia, thoracic horn and head horns). Genes possessing significantly higher expression in control animals were inferred to be those activated by Dsx, and those genes possessing significantly higher expression in *OtdsxC* dsRNA-injected animals were inferred to be those suppressed by Dsx.

Third, we analysed whether genes specifically identified as having sex-biased expression in control animals changed in expression levels across *OtdsxC* dsRNA-injected females and males across four tissues (brain, genitalia, thoracic horn and head horns). Significant differences between treatment groups and sexes were calculated by Wilcoxon tests and the Benjamini–Hochberg method was used to correct for multiple comparisons.

**Enrichment analyses.** We performed functional enrichment analyses to determine whether certain Gene Ontology terms were overrepresented among genes showing significant sex-biased expression or inferred to be regulated by Dsx. We characterized protein functions based on homology with described genes in the Gene Ontology database[60] using the programme Blast2GO (ref. 61), resulting in functional categories representing gene product properties (for example, contributions to cell structure, their involvement in molecular pathways or their role in a biological process). We used ErmineJ[62,63] to test whether each functional category was enriched for differentially expressed genes relative to all other genes using gene score resampling (scores were based on adjusted $P$ values) to generate a null distribution. We then compared our lists of functional categories that were enriched for either sex-biased or Dsx-mediated genes to the list of functional categories found to be enriched for Dsx targets as identified in *D. melanogaster*[9].

**Binding-site analysis.** Putative Dsx-binding sites in the *O. taurus* genome[55,64] were determined with the PoSSuM program[65]. The significance ($P < 0.005$) of a predicted Dsx binding site was determined by its similarity to the position weight matrix of *D. melanogaster* Dsx (CIS-BP database[66]). Custom Perl scripts were used to filter and associate predicted binding sites within the 5 kb regions upstream of the transcription start sites of genes that were non-overlapping with other genes. Because there is no established criterion to determine the distance from a promoter that an enhancer might be (it is difficult to experimentally associate enhancers with specific promoters regions), a 5 kb window was chosen to account for both the fact that most enhancers have been found close to promoter regions (within 500 bps), and that examples of enhancer that are quite distant from the promoter are also known in insects. Putative targets of Dsx were defined as genes that were both mediated by Dsx (that is, possessing differential expression between control and *dsx* knockdown animals) and possessing five or more significant predicted Dsx-binding sites. Targets were compared among sexes and tissues. Targets identified in this study were also compared with those found in a study that determined Dsx targets in the fruit fly *D. melanogaster* using an array of techniques including ChIP-Seq and DamID-Seq[9]. To make the Dsx targets identified in each study comparable, we first used BLASTx[67] to identify putative orthologues of the *D. melanogaster* Dsx targets in the *O. taurus* genome, defined as genes sharing at least 80% protein sequence identity between the two species.

**Candidate gene analysis.** To identify candidate genes and pathways targeted by Dsx, we compared *O. taurus* genes that were putative Dsx targets (as defined above) to their respective putative orthologues in other species using BLAST2GO (ref. 61), looking for genes that (i) had been previously identified as having a role in sex-specific development in *O. taurus*, (ii) had been implicated in sex-specific development in other species but are not yet known to have a role in *O. taurus* or (iii) that are not known to have a role in sex-specific development in *O. taurus* or other species, but contributed to pathways that appeared overrepresented in our data.

**Data availability.** Data generated in this study were deposited in NCBI's Gene Expression Omnibus[68] and are accessible through GEO Series accession number GSE87788 (https://www.ncbi.nlm.nih.gov/geo/query/acc.cgi?acc=GSE87788).

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

## Acknowledgements

We express our gratitude to the Indiana University Center for Genomics and Bioinformatics, in particular James Ford, Douglas Rusch and Aaron Buechlein. We also thank Teiya Kijimoto for designing the Ot*dsx*C construct, the members of the i5k pilot project for sequencing the O. taurus genome, and the members of the Moczek Lab for their general support and feedback on this manuscript. Funding for this study was provided by National Science Foundation grants IOS 1256689 and IOS 1120209 to A.P.M.

## Author contributions

C.C.L.-R., E.E.Z. and A.P.M. designed the experiments; C.C.L.-R. and E.E.Z. conducted the experiments (dsRNA injections, phenotyping, tissue dissection and RNA extraction); C.C.L.-R. designed and ran the data analysis pipeline; C.C.L.-R., E.E.Z. and A.P.M. interpreted the data; C.C.L.-R. wrote the main manuscript; C.C.L.-R. and E.E.Z. designed the figures; A.P.M. secured funding; C.C.L.-R., E.E.Z. and A.P.M. edited and approved the final manuscript.

## Additional information

**Competing financial interests:** The authors declare no competing financial interests.

