## [Peer Review File · Nature Communications]

Reviewers' Comments:

Reviewer #1 (Remarks to the Author)

This is an interesting paper with attempts at addressing how sexually dimorphic traits evolved as fueled by the functional promiscuity of a sex-determination gene, *doublesex*, by the combination of RNAi-mediated gene inactivation, RNA-seq and bioinformatic analyses. The paper needs clarifications in some points as described below.

1. line 175. Remove "the" between "determined" and "how".
2. The first character of protein names should be capitalized, followed by lower case letters throughout the manuscript; e.g., "Dsx" in Roman not "dsx" in italic (this rule must be also applied to other protein names).
3. *dsx* rather than *doublesex* should be used throughout the manuscript, except when *doublesex* appears for the first time in the text.
4. Unlike *Drosophila dsx*, *Onthophagus dsx* produces multiple female transcript forms (Figure 1). Do the authors have any evolutionary implications for this female-specific transcript diversification? What are possible impacts of having multiple female-specific transcripts on the quantitative/comparative analysis of differential *dsx* actions in two sexes?
5. Lines 388-397. In interpreting for the *dsx*-independent sex biases in gene expression, the authors cites Taylor 1992 to verify their statement for the presence of a *dsx*-independent mechanism. The *dsx*-independent mechanism suggested by Taylor is actually a *fru*-dependent mechanism, which was unknown in 1992. Because, at least in *D. melanogaster*, *fru* has no role in sex-determination outside the nervous system, it is unlikely that *fru* is involved in the *dsx*-independent biased expression observed in this study. Also note that no evidence has been obtained for hermaphrodite-mediated modifications of *fru* functions.
6. Lines 417-420. I do not understand why sexual selection can explain the authors' observation that female *dsx* isoforms have much more targets than the male counterpart does, because males are likely under stronger selective pressures than females in this species.

Reviewer #2 (Remarks to the Author)

High throughput sequencing methods have made it possible to compare gene expression between females and males on a genome-wide scale. Such analyses have revealed that sex-biased gene expression is abundant in many species. In addition, the range of sex-biased expression may vary greatly among tissues or developmental stages.

In this study the authors used high throughput sequencing, in conjunction with RNAi of *O. taurus* male and female *dsx* mRNA isoforms in larvae. They analysis gene expression changes across adult brains, genitalia, thoracic horns and head horns, was combined with an analysis of putative canonical *dsx* binding sites across the *O. taurus* genome.

Major points

The author's claim that *dsx* regulates sex-biased expression predominantly in males, that *dsx*'s target repertoires are highly sex- and tissue-specific, and that *dsx* can exercise its regulatory role via two distinct mechanisms: as a sex-specific modulator by regulating strictly sex-specific targets, or as a switch by regulating the same genes in males and females in opposite directions. These are not new revelations about *dsx* regulation of gene expression. Although this study produces some solid bioinformatic data, with the lack of any functional studies on *dsx*-'targets', it's not clear that this study really uncovers anything new about the evolutionary forces that govern the evolution of sex-biased genes. It seems more suitable for a journal like *Genome Biology and Evolution*.

Minor points

The data in Figure 1 has all been published before in Kijimoto et al. (2012), and should be removed or added to supplemental methods. It would also be helpful to see the intersexual phenotypes of the external morphological structures.

Reviewer #3 (Remarks to the Author)

The DMRT transcription factors are widely conserved and are implicated in sex determination and/or differentiation in those species. The authors have identified putative Dsx target genes required for sex dimorphism in the horned beetle (*Onthophagus Taurus*). Males of this species have highly sexually dimorphic thoracic and head horns, making it a very interesting model for the study of the important DMRT family. The authors used RNA-seq to study the transcriptomic response to RNAi knockdown of dsx transcripts in four tissues. Additionally, the authors took advantage of predicted Dsx binding sites to make inference about direct versus indirect target genes (although this could be expanded). The most interesting finding is that like in *Drosophila*, the function of Dsx appears to be highly tissue and sex-specific. This is a nice confirmation of previous genome-wide suggestions that there are interactions of Dsx with other regulators in multiple tissues to upregulate or downregulate genes. While I find this work very interesting, I found the level of analysis superficial and very poorly described. There are a number of questions about the RNAseq datasets that must be resolved and some more extensive comparisons of both the types of genes regulated by Dsx and context dependence in Beetles versus other organisms should be included (the authors will get much more credit too). This is relatively straightforward, given the existing high-throughput datasets for Dsx genomics in other organisms.

Major:

1. I will need access to the GEO (or other repository) entry for this dataset.
2. Many details on the method, including the how many replicates the authors performed, need to be provided and discussed. If there are replicates, do they have high correlation for all tissues? The complexity of the tissue type or dissection could influence the results and conclusions.
3. Data handling decisions are poorly described and often appear arbitrary. For example, what is the justification for 5kb windows and a 5 Dsx binding motifs cutoff?
4. The single sentence description of "candidate gene analysis" is another example of a case where it would be virtually impossible for another group to replicate the analysis after reading the methods.
5. What is the knockdown efficiency? This should be self-reporting in the dataset (unless dsx transcripts are too rare). This should be reported.
6. It would be useful to say something more about the classes of genes that show either sex-biased or Dsx-dependent expression. GO terms might be useful. This might need to be done by using terms from orthologs in other species, but it would still be interesting to know if these genes show sex-bias in other species, if they encode enzymes or transcription factors, etc. It should be straightforward to enrich the supplementary tables (The gene IDs alone are not very useful) to characterize both sex-biased expression and Dsx regulation.
7. The similarities and differences between the *Drosophila* and Beetle dsx target genes is not analyzed or even clearly acknowledged in the manuscript. A comparison of the type and behavior of putative Dsx targets in HT data on Dsx (VanDoren/Oliver and Baker for fly, and Zarkower for mouse) is needed. The Clough et al 2014 paper came to the same "context dependent role of Dsx" conclusions as the authors, so this would be an especially interesting compare and contrast case. A more complete comparative analysis will make this a valuable paper for a larger audience.
8. I had difficulty understanding the "simplifying" Fig. 3 and I think many readers will just ultimately give up (the text was more clear). The terms dsx-M and dsx-F should refer to isoforms, not RNAi, which will confuse people. Expression is biased, not genes. The legend states that "female dsx isoforms only modulate sex-biased gene expression in a single female tissue", but the green check marks in 3C are the same for both male- and female-biased responses. I get that only

the head horn is affected in females (but both male- and female-biased ones?). Substantial changes in presentation are needed here. Maybe a heatmap to show the complex non-overlapping pattern?

9. The head horns have more consistent expressional levels among different tissues in both female and male. The values show little contrast between sexes (female/male) or between treatments (control/knockdown), as compared to other tissues. Are most of the DE genes of head horns marginally significant? Again, a different form of presentation showing (as appropriate) the number of genes, the magnitude, and the significance would be helpful.

Minor:

(1) While flies and beetles regulate dsx by splicing, this is not true for all insects. It would be nice to have a short introduction to sex determination (primary signals, transformer, etc) in beetles.

(2) Why is a paucity of sex-biased expression in the brain surprising?

(3) Results and figure legends are often redundant. Many sentences seems to be copied and pasted with very minor modifications.

(4) The authors used "dsx-M" or "dsx-F" for shorthand in several places, but this is confusing as dsx-M and dsx-F usually refer to transcripts. Indicating "dsx-RNAi" and the sex would be clearer.

(5) I do not find the speculative model in Figure 4 very useful, and would prefer seeing more data directly comparing Dsx modes in beetles with orthologs in other organisms.

RESPONSE TO REVIEWER COMMENTS

Reviewer comments are in *gray italics* and our responses are in black, normal type.

Reviewer #1 (Remarks to the Author):

1. line 175. Remove "the" between "determined" and "how".

Thank you for pointing out this typo; it has been corrected.

2. The first character of protein names should be capitalized, followed by lower case letters throughout the manuscript; e.g., "Dsx" in Roman not "dsx" in italic (this rule must be also applied to other protein names).

Thank you for pointing out this inconsistency; it has been corrected throughout the text.

3. dsx rather than doublesex should be used throughout the manuscript, except when doublesex appears for the first time in the text.

Thank you for pointing out this error; it has been corrected throughout the text

4. Unlike Drosophila dsx, Onthophagus dsx produces multiple female transcript forms (Figure 1). Do the authors have any evolutionary implications for this female-specific transcript diversification? What are possible impacts of having multiple female-specific transcripts on the quantitative/comparative analysis of differential dsx actions in two sexes?

We agree with the reviewer that the pattern wherein insects possess one male *dsx* isoform, yet variable numbers of female *dsx* isoforms, is intriguing and may have evolutionary implications. It is possible that the evolution of the female splice sites, and the resulting female isoforms, is akin to gene duplication followed by neofunctionalization, where new isoforms have greater freedom to diversify and confer new traits to females (only). However, there is limited empirical data to support this notion; even in the case of *dsx* underlying female-limited mimicry in *Papilio*, while female *dsx* was more highly expressed in mimetic females, all female isoforms were highly expressed (i.e., it is not clear that having more than one female isoform contributes to phenotypic diversity). Given the paucity of data linking different female isoforms to divergent phenotypes, we would prefer not to expand on this topic. However, if the reviewer or editor feels strongly about that we should, we will attempt to do so.

With regards to the effects of multiple female isoforms on the comparisons of male- and female actions among sexes, we believe that our approach addresses this concern; our RNAi was designed to knock down all female isoforms. Because our study focused on the differences between male and female isoforms, and not on the differences between female isoforms, we do not believe that we have analyzed or interpreted our data inaccurately.

5. Lines 388-397. In interpreting for the dsx-independent sex biases in gene expression, the authors cites Taylor 1992 to verify their statement for the presence of a dsx-independent mechanism. The dsx-independent mechanism suggested by Taylor is actually a fru-dependent mechanism, which was unknown in 1992. Because, at least in D. melanogaster, fru has no role in sex-determination outside the nervous system, it is unlikely that fru is involved in the dsx-independent biased expression observed in this study. Also note that no evidence has been obtained for hermaphrodite-mediated modifications of fru functions.

We thank the reviewer for highlighting details regarding the Taylor study. Our purpose was not to suggest that fruitless specifically mediates female-biased gene expression (indeed, the paper focuses on a male-biased trait), but to simply point out that there are *Dsx*-independent mechanisms for generating sex-biased expression. Further, even if *fru* operates exclusively in the nervous system, it must be able to also influence tissues outside the nervous system; the *dsx*-independent traits in the Taylor study was a muscle that was dependent on motoneuronal

innervation. Thus, because the *O. taurus* pupa is enervated, *fru* may in fact be responsible for some of the female-biased expression in the non-neuronal tissues of our study. However, the use of this example was apparently misleading, so we have removed it from the text, which does not alter the meaning or impact of the paragraph.

6. Lines 417-420. I do not understand why sexual selection can explain the authors' observation that female dsx isoforms have much more targets than the male counterpart does, because males are likely under stronger selective pressures than females in this species.

We were actually not arguing that the female isoforms have more Dsx targets in female head horns than the male isoform does in male head horns (indeed, males have far more Dsx targets in head horns than do females), rather, we were arguing that the female isoforms have more targets in the female head horns than they do in other female tissues. We have rewritten the sentence to better reflect this argument (changes in bold),

“We would predict that this trend – wherein female Dsx isoforms have acquired disproportionately more genetic targets **in a sexually selected trait than in other female traits** – might be observed in other traits that are under intense sexual selection in other insect taxa, such as the eye stalks of diopsid flies or mandibles in stag beetles.”

Further, although males may be under stronger sexual selection to grow larger horns, females may be under natural selection to suppress horns. Some studies have found that horns inhibit maneuverability in tunnels, which is essential for females since they shoulder most of the burden of constructing tunnels and broodballs (Madewell and Moczek, 2006). Genes underlying horn growth might then be sexually antagonistic, and under strong selective pressure to decouple female and male expression patterns (Rice and Chippindale 2001). Sex-specific gene expression is one way of circumventing these constraints (and resulting losses in fitness), such that the evolution of novel Dsx targets in female head horns might be rapid relative to other female traits that are not underlain by sexually antagonistic genes.

If the reviewer or editor believe that any elements of this argument should be included in the manuscript, we would be happy to add them.

Reviewer #2 (Remarks to the Author):

High throughput sequencing methods have made it possible to compare gene expression between females and males on a genome-wide scale. Such analyses have revealed that sex-biased gene expression is abundant in many species. In addition, the range of sex-biased expression may vary greatly among tissues or developmental stages.

*In this study the authors used high throughput sequencing, in conjunction with RNAi of *O. taurus* male and female dsx mRNA isoforms in larvae. They analysis gene expression changes across adult brains, genitalia, thoracic horns and head horns, was combined with an analysis of putative canonical dsx binding sites across the *O. taurus* genome.*

Major points

The author's claim that dsx regulates sex-biased expression predominantly in males, that dsx's target repertoires are highly sex- and tissue-specific, and that dsx can exercise its regulatory role via two distinct mechanisms: as a sex-specific modulator by regulating strictly sex-specific targets, or as a switch by regulating the same genes in males and females in opposite directions. These are not new revelations about dsx regulation of gene expression. Although this study produces some solid bioinformatic data, with the lack of any functional studies on dsx-

'targets', it's not clear that this study really uncovers anything new about the evolutionary forces that govern the evolution of sex-biased genes. It seems more suitable for a journal like Genome Biology and Evolution.

We respectfully disagree with the sentiment of the reviewer; while our study corroborates some previously described genetic targets of *dsx*, our study is still the most comprehensive assessment, to date, of *Dsx*'s targets across sexes in an insect. This had led to insights that cannot be gleaned from previous studies including (i) the ease with which *Dsx* can assume novel functions in novel traits, (ii) that there is an asymmetry in the size of *Dsx*'s target repertoire between sexes, and (iii) that whether *Dsx* operates as a sex-specific modulator by regulating strictly sex-specific targets, or as a switch by regulating the same genes in males and females in opposite directions depends on whether a trait is sexually selected. These findings are unprecedented and enhance our current understanding of the role of *Dsx* in the sexual development of insects.

Minor points

The data in Figure 1 has all been published before in Kijimoto et al. (2012), and should be removed or added to supplemental methods. It would also be helpful to see the intersexual phenotypes of the external morphological structures.

We appreciate that the reviewer has read Kijimoto et al. 2012, but disagree that it not useful to show that our new data (not borrowed from Kijimoto et al. 2012) corroborates the findings from Kijimoto et al. 2012. Indeed, this publication inspired the current manuscript, and we would not have proceeded had we not been able to faithfully reproduce these phenotypes. Further, the intersexual phenotypes requested by the reviewer are in fact portrayed in Figure 1.

Reviewer #3 (Remarks to the Author):

The DMRT transcription factors are widely conserved and are implicated in sex determination and/or differentiation in those species. The authors have identified putative Dsx target genes required for sex dimorphism in the horned beetle (Onthophagus Taurus). Males of this species have highly sexually dimorphic thoracic and head horns, making it a very interesting model for the study of the important DMRT family. The authors used RNA-seq to study the transcriptomic response to RNAi knockdown of dsx transcripts in four tissues. Additionally, the authors took advantage of predicted Dsx binding sites to make inference about direct versus indirect target genes (although this could be expanded). The most interesting finding is that like in Drosophila, the function of Dsx appears to be highly tissue and sex-specific. This is a nice confirmation of previous genome-wide suggestions that there are interactions of Dsx with other regulators in multiple tissues to upregulate or downregulate genes. While I find this work very interesting, I found the level of analysis superficial and very poorly described. There are a number of questions about the RNAseq datasets that must be resolved and some more extensive comparisons of both the types of genes regulated by Dsx and context dependence in Beetles versus other organisms should be included (the authors will get much more credit too). This is relatively straightforward, given the existing high-throughput datasets for Dsx genomics in other organisms.

Major:

1. I will need access to the GEO (or other repository) entry for this dataset.

We have submitted our dataset to GEO and it's accession number is: GSE87788

The reviewer access link is,

<https://www.ncbi.nlm.nih.gov/geo/query/acc.cgi?token=cdupykgannitpab&acc=GSE87788>

2. Many details on the method, including the how many replicates the authors performed, need to be provided and discussed. If there are replicates, do they have high correlation for all tissues? The complexity of the tissue type or dissection could influence the results and conclusions.

In the original manuscript we did include a statement about how many replicates there were on New Line 138-139:

“Six male and six female pupae weighing over 125 mg from each group were used for tissue dissection and total RNA extraction.”

And also New Line 153-154:

“A total of 96 (2 treatments x 2 sexes x 4 tissues x 6 biological replicates) RNA Stranded RNA sequencing libraries were constructed...”

In our revisions, we have also added Supplementary Material S2 that includes a correlational heatmap and PCA plot of our samples, along with Supplementary Table T1 summarizing pairwise Pearson’s correlation coefficients between biological replicates. We hope that the reviewers find this sufficient, however we are happy to include further analyses if necessary.

3. Data handling decisions are poorly described and often appear arbitrary. For example, what is the justification for 5kb windows and a 5 Dsx binding motifs cutoff?

Because there is no widely accepted way to determine the distance from a promoter that an enhancer might be (it is difficult to associate enhancers with specific promoters), we decided to use a 5kb window to respect the fact that most regulatory elements are near to promoters (within 500bps), but also that insects can have enhancer elements that are quite distant from the promoter. The reason we chose to only include genes that had at least 5 significant Dsx binding sites was to reduce the number of genes that had one or a few binding sites just by chance.

4. The single sentence description of "candidate gene analysis" is another example of a case where it would be virtually impossible for another group to replicate the analysis after reading the methods.

We agree that description of this analysis would have benefited from more detail. No strict rules were used to delineate a candidate gene from non-candidate genes. Instead, we searched for genes that might fit at least one of three different categories, those that (i) had been previously identified as having a role in sex-specific development in *O. taurus*, (ii) had been implicated in sex-specific development in other species but are not yet known to have a role in *O. taurus*, or (iii) that are not known to have a role in sex-specific development in *O. taurus* or other species, but contributed to pathways that appeared overrepresented in our data. This approach has now been explained on New Lines 234-238.

5. What is the knockdown efficiency? This should be self-reporting in the dataset (unless dsx transcripts are too rare). This should be reported.

We thank the reviewer for pointing out this important oversight. We have now reported the dsxRNAi knockdown efficiency in Supplementary Material S1. The only instance in which dsxRNAi failed to significantly reduce levels of dsx expression was in the case of female brains; while our results pertaining to female brains should thus be taken with a grain of salt, there are reasons why this knockdown may have occurred without measurably reducing dsx expression levels. Specifically, (i) levels of expression in the brain are already very low, so changes might

be hard to detect, and (ii) we know from prior experience with RNAi in *Onthophagus* as well as *Tribolium* that obvious knockdown phenotypes can be achieved without detectable reductions in gene expression levels, for example due to compensatory expression.

6. It would be useful to say something more about the classes of genes that show either sex-biased or Dsx-dependent expression. GO terms might be useful. This might need to be done by using terms from orthologs in other species, but it would still be interesting to know if these genes show sex-bias in other species, if they encode enzymes or transcription factors, etc. It should be straightforward to enrich the supplementary tables (The gene IDs alone are not very useful) to characterize both sex-biased expression and Dsx regulation.

We thank the reviewer for this useful suggestion. We have now performed a gene ontology enrichment analysis, included it in our methods, results and discussion, and additionally have provided a new Supplementary Material Excel Workbook (Supplementary Table T3) that contains our findings, in detail. We have also compared the gene ontology terms identified in our study with those found in a study of Dsx targets in *D. melanogaster* (Clough et al. 2014), and those results are provided in a new Supplementary Material Excel Workbook (Supplementary Table T4).

7. The similarities and differences between the Drosophila and Beetle dsx target genes is not analyzed or even clearly acknowledged in the manuscript. A comparison of the type and behavior of putative Dsx targets in HT data on Dsx (VanDoren/Oliver and Baker for fly, and Zarkower for mouse) is needed. The Clough et al 2014 paper came to the same "context dependent role of Dsx" conclusions as the authors, so this would be an especially interesting compare and contrast case. A more complete comparative analysis will make this a valuable paper for a larger audience.

We agree with the reviewer that a comparison between *Drosophila* and beetle Dsx target genes would make this paper more valuable for a larger audience. Before comparing the 3,717 genes identified as Dsx targets in the Clough paper to the Dsx targets identified by our analysis, we first attempted to account for differences in the levels of annotation that have been accomplished in the *O. taurus* and *D. melanogaster* genomes; i.e., it is possible that some of the genes annotated in *D. melanogaster* may not yet be annotated in *O. taurus*, making these lists hard to compare. To do this, we first used BLAST to find potential *O. taurus* orthologs of the 3,717 *Drosophila* genes identified in the Clough paper. This process yielded 603 *Drosophila* Dsx target genes that could be compared to putative *Onthophagus* target genes. We then compared these 603 to the 12,776 total potential Dsx targets generated by our PoSSuM binding analysis, and found that 444, or approximately 74%, of the Dsx targets identified in the Clough paper (that had orthologs in *O. taurus*) were also identified as Dsx targets in our study. We have written the methods and results into New Lines 225-230, New Lines 387-393 and the results are also provided in Supplementary Table T6.

8. I had difficulty understanding the "simplifying" Fig. 3 and I think many readers will just ultimately give up (the text was more clear). The terms dsx-M and dsx-F should refer to isoforms, not RNAi, which will confuse people. Expression is biased, not genes. The legend states that "female dsx isoforms only modulate sex-biased gene expression in a single female tissue", but the green check marks in 3C are the same for both male- and female-biased responses. I get that only the head horn is affected in females (but both male- and female-biased ones?). Substantial changes in presentation are needed here. Maybe a heatmap to show the complex non-overlapping pattern?

We agree with the reviewer that "dsx-M" and "dsx-F" should refer to isoforms, not RNAi, and have incorporated this change into the figure. We also agree with the reviewer that expression is biased, not genes, and have modified the caption to read "genes with female-biased

expression” or “genes with male-biased expression” to remedy this error (we have also eradicated this misuse throughout the rest of the manuscript). Finally, we also agree with the reviewer that the initial rendition of the figure was confusing, and have modified the figure greatly to generate one that is easier to understand. We hope that the reviewer finds the new Figure 3 helpful, but we are open to further modification if necessary.

Heatmaps were generated (provided as an additional document in our response to reviewers) in order to determine whether this was a better way of displaying the pattern observed in our data. We do not think that these heatmaps portray the patterns in the data more easily than the existing boxplots, however, if the reviewer feels strongly that they do, we are more than happy to include them in the final version of the manuscript, either as a main or supplementary figure.

9. The head horns have more consistent expressional levels among different tissues in both female and male. The values show little contrast between sexes (female/male) or between treatments (control/knockdown), as compared to other tissues. Are most of the DE genes of head horns marginally significant? Again, a different form of presentation showing (as appropriate) the number of genes, the magnitude, and the significance would be helpful.

We apologize but we are not sure we fully understand what the reviewer’s comment is getting at. All genes that were used to compare among groups in Figure 3 are significantly differentially expressed, as assessed between control males and control females (described in New Lines 178-181), using negative binomial models followed by corrections for multiple comparisons. Thus, none of the genes in any tissue used in Figure 3 were marginally significant. The magnitude of expression of the genes is represented as Log (FPKM + 1). The number of sex-biased genes among tissues is presented in Supplementary Figure 3.

Minor:

(1) While flies and beetles regulate dsx by splicing, this is not true for all insects. It would be nice to have a short introduction to sex determination (primary signals, transformer, etc) in beetles.

We thank the reviewer for making this suggestion. We have now included a brief synopsis of the role of *transformer* in *dsx* splicing in beetles (New Lines 49-56), although that role seems ambiguous at this point because *transformer* was not found in *Onthophagus taurus* or three other beetle species aside from *Tribolium*. We actually feel that discussing the signals upstream of *dsx* is out-of-place in our manuscript given that we have provided no data that enhances the current understanding of those signals in beetles (which, as mentioned, is limited), but if the reviewers and editor agree that this discussion should be had, we will include the synopsis in the final manuscript.

(2) Why is a paucity of sex-biased expression in the brain surprising?

We have removed “unexpected” from that sentence – we agree that this was phrasing was unnecessary.

(3) Results and figure legends are often redundant. Many sentences seems to be copied and pasted with very minor modifications.

We have revised the manuscript so that it contains less redundancy.

(4) The authors used "dsx-M" or "dsx-F" for shorthand in several places, but this is confusing as dsx-M and dsx-F usually refer to transcripts. Indicating "dsx-RNAi" and the sex would be clearer.

We agree strongly with the reviewer that the use of dsx-M and dsx-F is confusing, and we have removed all instances of these terms from the manuscript.

(5) I do not find the speculative model in Figure 4 very useful, and would prefer seeing more data directly comparing Dsx modes in beetles with orthologs in other organisms.

While we respectfully disagree that the model in Figure 4 is not useful, we have rewritten the caption to make it clearer. The focus of the comparisons in Figure 4 are to show that the modes of Dsx's actions differ based on both sex and tissue, which we believe is an important contribution to our general understanding of Dsx. Further, because the figure includes an ancestral state (basal arthropods), tissues that are homologous among insects, and a tissue that is novel among insects, it also provides a hypothesis about the evolution of Dsx's functions between sexes and among tissues. This model is of course speculative and requires further studies to support, but it is nevertheless compelling.

We would also enjoy seeing more data directly comparing the mode of Dsx between sexes and among tissues in a comparative framework (i.e., directly comparing beetles and other organisms), however, to our knowledge, modes of Dsx between sexes and among tissues are

Reviewers' Comments:

Reviewer #1 (Remarks to the Author)

The issues raised in the first round of review have been addressed and I think the manuscript is now acceptable for publication.

Reviewer #3 (Remarks to the Author)

The manuscript is improved by the cross-species comparison and the inclusion of more information on what types of genes are potentially regulated by Dsx. The GEO entry is mostly in order. A few minor points:

I think it would be nice to include the information on suppression of horn formation and implications for tunneling as suggested in the response to reviewer 1.

Reviewer 2 made a point of the lack of novelty in terms of dsx regulation, but I like seeing a nice example from another species and find the work valuable.

Was there any overlap with mouse DMRT1 targets?

Change the confusing sample names (dsxF and dsxM) in GEO GSE87788 to match the descriptions in the manuscript.

State that cutoffs are arbitrary and give justification for the reader, not just the reviewer.

"biased genes" still appears in figure 1B. It should be "biased expression".

REVIEWERS' COMMENTS:

Reviewer #1 (Remarks to the Author):

The issues raised in the first round of review have been addressed and I think the manuscript is now acceptable for publication.

We are glad that we have addressed the reviewer's comments from the first round of reviews.

Reviewer #3 (Remarks to the Author):

The manuscript is improved by the cross-species comparison and the inclusion of more information on what types of genes are potentially regulated by Dsx. The GEO entry is mostly in order. A few minor points:

I think it would be nice to include the information on suppression of horn formation and implications for tunneling as suggested in the response to reviewer 1.

We agree and have now included the discussion of sexual antagonism in the discussion section.

Reviewer 2 made a point of the lack of novelty in terms of dsx regulation, but I like seeing a nice example from another species and find the work valuable.

We agree completely, and appreciate your support.

Was there any overlap with mouse DMRT1 targets?

We decided not include a comparison of our Dsx targets with mouse DMRT1 targets. Given that our results suggest that – in spite of the fact that the role of DMRTs in sexual differentiation is highly conserved – the target repertoires of Dsx are highly labile even between sexes and among tissues, we did not expect that many targets would be shared between the two phylogenetically distant groups. Further, an additional comparison would require a significant amount of space that might compromise more informative sections of the manuscript.

Change the confusing sample names (dsxF and dsxM) in GEO GSE87788 to match the descriptions in the manuscript.

We have changed the GEO sample names. For example, “dsxF” has now become “dsxRNAi_F” to indicate that “dsx” refers to the RNAi used and the “F” refers to the sex treated.

State that cutoffs are arbitrary and give justification for the reader, not just the reviewer.

December 13, 2016

We have included a statement as to the arbitrary nature of the 5kb cutoff for the binding site analysis in our methods.

"biased genes" still appears in figure 1B. It should be "biased expression".

Thank you for pointing this out. We were not able to find the instance of "biased genes" Figure 1B, but did in Figure 2B and 2C, as well as instances of this misuse in the legend of Figures 2 and 4. We have now corrected all instances of "biased genes".